# Optimal Synthesis of Novel Phosphonic Acid Modified Diatomite Adsorbents for Effective Removal of Uranium(VI) Ions from Aqueous Solutions

**DOI:** 10.3390/ma16155263

**Published:** 2023-07-26

**Authors:** Natalia Kobylinska, Oksana Dudarko, Agnieszka Gładysz-Płaska, Valentyn A. Tertykh, Marek Majdan

**Affiliations:** 1A.V. Dumansky Institute of Colloid and Water Chemistry, National Academy of Science of Ukraine, 42, Akad. Vernadskogo Blvd., 03142 Kyiv, Ukraine; 2Chuiko Institute of Surface Chemistry, National Academy of Science of Ukraine, 17 General Naumov Str., 03164 Kyiv, Ukraine; 3Institute of Chemical Sciences, Faculty of Chemistry, Maria Curie-Sklodowska University, M. Curie-Sklodowska Sq. 2, 20-031 Lublin, Poland

**Keywords:** natural diatomite, covalently immobilized groups, silanization, surface modification, phosphonic acid, radionuclides, water purification, wastewaters

## Abstract

The authors synthesized a series of functionalized diatomite-based materials and assessed their U(VI) removal performance. Phosphor-derivative-modified diatomite adsorbents were synthesized by the three-route procedures: polymerisation (*DIT-Vin-PA_in_*), covalent (*DIT-Vin-PA_cov_*), and non-covalent (*DIT-PA*) immobilization of the functional groups. The effects of the diatomite modification have been studied using powder XRD, solid state NMR, FTIR spectroscopy, electronic microscopy, EDX, acid–base titrations, etc. The maximum adsorption capacities of *DIT-Vin-PA_cov_*, *DIT-PA*, and *DIT-Vin-PA_in_* samples were 294.3 mg/g, 253.8 mg/g, and 315.9 mg/g, respectively, at pH_0_ = 9.0. The adsorption amount of U(VI) ions using the prepared *DIT-Vin-PA_in_* was 95.63%, which is higher compared with that of the natural diatomite at the same concentration. The adsorption studies demonstrated that the phosphonic and hydroxyl groups on the surface of the diatomite played pivotal roles in the U(VI) adsorption. The U(VI) ions as a “*hard*” Lewis acid could easily form bonds with the “*hard*” donor P-containing ligands, so that the as-prepared *DIT-Vin-PA_in_* sample had excellent adsorption properties. The monolayer adsorption of the analyte on the surface of the raw diatomite and *DIT-PA* was observed. It was found from the thermodynamic parameters that the uptake of the U(VI) ions by the obtained adsorbents was a spontaneous process with an endothermic effect. Findings of the present work highlight the potential for using modified diatomite as effective and reusable adsorbents for the extraction of U(VI) in the waste, river, and tap waters with satisfactory results.

## 1. Introduction

The intensive development of modern industries, such as nuclear energy, human health, and the remediation of ecosystems, is subject to serious ecological dangers as a result of radioactive and chemical pollutions due to the release of radionuclides into the environment [1]. The radionuclide pollution presents a worldwide problem. The accumulation of radionuclides in the environment affects water quality, food safety, and human health. Among the elements used in nuclear energy, uranium is one of the most hazardous heavy metal ions due to its high toxicity as well as its radioactivity (which should not exceed 1.0 Bq/L) and long half-life [2]. Natural uranium is a mixture of three isotopes (^234^U, ^235^U, and ^238^U), but isotope ^235^U is useful as a fuel in power plants and weapons. Special attention is paid to the consequences of the catastrophic nuclear accidents at the Chernobyl (in 1986) and Fukushima Daiichi (in 2011) Nuclear Plants [3], the occupation of the highest nuclear power objects in Europe (Chernobyl and Zaporizhzhya) as a result of the war in Ukraine (during the years 2022–2023), etc. The purification of environmental waters contaminated by uranium compounds is a topical problem in these technogenic catastrophes. The uranium concentrations in the groundwaters were not higher than the Maximum Contaminant Level (MCL) of 30 μg/L as recommended by United States Environmental Protection Agency (U.S. EPA) nor the Maximum Contaminant Level Goal of zero [4], and the WHO has established the MCL for uranium to be at 10 µg/L in drinking water [5]. Thus, the extraction of U(VI) from polluted solutions is of ultimate importance for human health and environmental safety.

Frequently, various physical and chemical methods (chemical precipitation, membrane separation [5], ion exchange, electrodialysis, photocatalysis, microbial reduction [6] and adsorption, etc.) are applied for the remediation of uranium from environmental water. Among these processes, adsorption is the most convenient and widely used method due to its low cost and convenient operation. The most critical point for adsorption is the preparation of efficient adsorbents. For this purpose, various inorganic and polymeric materials have been used [7] and reviewed [8,9].

A variety of adsorbents with different ligands have been designed for the exaction of radionuclides, such as graphene oxide, layered double hydroxide(LDH) [9,10], nanoscale zero-valent iron and direct-reduced iron (sponge iron) [11], magnetite [8,12,13,14], resin [15], goethite [16,17], europium hydroxide [18], self-assembled monolayers on nanocomposites [19,20,21], silica [22] and polymeric derivatives [23,24,25], carbon nanotubes, modified biowaste [26], natural minerals [27], and nano-sized filtration membranes [28]. The surface of these materials can be modified with functional groups or molecules, such as natural polymer (chitin or chitosane) [21], aminoacids [18], acidic [29] and chelating [10] groups, amidoxime-functionalized macroporous fibrous polymeric [23], poly(acrylic acid) [24], and activated cellulose and silica grafted cellulose [30]. Among them, the carboxyl groups showed a good affinity for all radionuclides without selectivity. It is known that U(VI) is considered as a Lewis hard acid, leading to the high affinity for P-containing groups [31]. For example, phosphate-functionalized materials, such as phosphorylated chitosan graphene oxide and carbon aerogel foam, showed a higher adsorption performance for U(VI) than those of the non-factionalized matrixes [32,33,34]. Zeng et al. [35] prepared phosphonate-functionalized polystyrene microspheres with controllable zeta potentials for the efficient capture of U(VI) from an aqueous solution. Chelated P-containing cation exchange resins under the trade names Lewatit OC 1060 (Bayer), Duolite ES 467 (Rohm & Haas), Purolite S940 and 950 (Purolite Intl), and Chelite P (Serva) are commercially available and have long been used for uranium pre-concentration and thorium and other trace elements from natural and industrial waters [36,37]. Commercialized chelating ion exchange resins have been widely used [38], including those mainly containing iminodiacetate (e.g., Lewatit TP 207 and Purolite S-930) and aminomethylphosphonate (e.g., Lewatit TP 260, Purolite S-940, and Purolite S-950) groups [39]. Thus, P-derivative adsorbents exhibit a strong affinity for radionuclides and have promising selectivity in their application to U(VI) removal for environmental protection.

The morphologies, structure, and surface charge of the adsorption materials have obvious influences on their adsorption properties. Although nanostructured materials have higher specific surface areas than micro-scale materials, they tend to aggregate, which decreases the effective surface states and leads to a reduction in their adsorption capacity for pollutants [8]. Micro-sized materials, constructed by integrating nanomaterials into the micro-scale materials or composites, have superior properties due to their developed surface area without aggregation properties and their good stability. The diatomaceous earth or diatomite is a low-cost natural micro/nanostructured material derived from sedimentary silica with the residue of dead algae; as a result, the mineral has cylindrical and plate morphologies with well-developed hierarchical porosity. Natural diatomite shows less efficiency for the removal of organic and inorganic pollutants [40]. Therefore, it is very important to modify natural diatomite via the purification and activation technique in an attempt to optimize the functionalization process and improve the structurally enhanced absorption of the toxic compounds [41]. Various modification approaches of diatomite have been described, including thermal treatment [42], acid treatment [43], and chemical modification of the surface [39,44]. Compared with other methods, the acid treatment methods can be readily applied to the large-scale production of modified diatomite because the simple technology to eliminate impurities, such as carbonate groups for implementing hydroxyl ions, and the low cost combine to save time and resources. Nevertheless, the fixation of chemical modifiers through the physical adsorption alone is associated with the risk of polluting the treated water with the leaching molecules of the modifier precursor. The covalent bonding of the functional groups enables the stable immobilization of the functional layer and allows reusability. The natural and surfactant-modified diatomite were studied to remove uranium ions from the aqueous solutions. It was established that the maximum adsorption capacities of the natural and the modified diatomite towards U(VI) were 25.63 mol/g and 667.40 mol/g, respectively [45]. Summing up, the use of diatomite and the improvement of the adsorption efficiency by modifying the surface provides an inexpensive and environmentally friendly approach to water-treatment technologies. However, to the best of our knowledge, there is no report in the literature of the use of phosphonic acid modified diatomite for uranium removal.

The present study is devoted to the synthesis and characterization of the structural and adsorption properties of diatomite with covalently and non-covalently immobilized phosphonic acidic groups. The major modified mechanisms between organosilanes and diatomite were assessed using characterization methods such as X-ray diffraction (XRD), solid state nuclear magnetic resonance (NMR), scanning electron microscopy coupled with energy-dispersive X-ray (EDX), Fourier-transform infrared (FTIR) spectroscopy, and pH-metric and conductometric titrations. The increase in the selectivity towards U(VI) ions compared with the initial natural diatomite was determined by the bench experiments and isotherm conditions. The adsorption isotherms have been analysed by the Langmuir, Freundlich, and Dubinin-Raduschkevich models. The thermodynamic parameters (ΔG°, ΔH°, and ΔS°) have also been evaluated by studying the influence of the temperature variation on the reaction. The specific ideas of this study are as follows: (1) develop innocuous, low-cost, and efficient adsorbents with high adsorption capacity for U(VI) ions in a real water medium; (2) evaluate the sustainable use and static U(VI) adsorption performance of the three developed P-containing adsorbents to adsorb U(VI) in real environmental water and wastewater; and (3) explore the U(VI) adsorption mechanism of the three adsorbents in raw actual environmental water and wastewaters.

## 2. Materials and Methods

### 2.1. Reagents

The reagents used in the experiments were triethylamine (Et_3_N, ≥ 99.5%, Sigma-Aldrich, St. Louis, MO, USA), toluene (anhydrous, Sigma-Aldrich), EtOH (pure ≥ 96.0%, anhydrous), trimethoxy(vinyl)silane (pure ≥ 97.0%, Sigma-Aldrich), phosphonic acid (H_3_PO_3_, crystalline, Alfa Aesar, 85 wt. %, Haverhill, MA, USA), azobisisobutyronitrile (AIBN, Sigma-Aldrich 98%), UO_2_(CH_3_COO)_2_ · 2H_2_O (czda, Chmes, Baton Rouge, LA, USA).

The reagents were analytical reagent grade. Stock solutions (1000 mg/L) of Ca^2+^, Mg^2+^, Mn^2+^, Fe^3+^, Cu^2+^, Na^+^, Co^2+^, Zn^2+^, Cd^2+^, Ni^2+^, CO_3_^2−^/HCO_3_^−^, and SCN^−^ were prepared by direct dissolution of proper amounts of Ca(NO_3_)_2_·4H_2_O, MgSO_4_·7H_2_O, MnCl_2_·4H_2_O, FeCl_3_·6H_2_O, CuSO_4_·5H_2_O, NaNO_3_, CoCl_2_ ·6H_2_O, Zn(NO_3_)_2_ 6H_2_O, CdCl_2_, NiCl_2_·6H_2_O, Na_2_CO_3_/NaHCO_3_, and KSCN. The standard solutions used for the calibration were prepared before use by dilution of the stock solution with HNO_3_.

### 2.2. Materials

Raw natural diatomite sample was obtained by Mikrosilika Trade from Stalowa Wola (Poland) minerals. The raw mineral was acidic-purified in 10 wt.% sulfuric acid at room temperature for 4 h. Then, the mineral was filtered, and the residue was washed with distilled water several times until the pH value reached 5.8 (distillate water). The acid-treated diatomite was dried during overnight at 100 °C.

### 2.3. Synthesis of Adsorbents

Synthesis of diatomite with covalent-grafted vinyl groups (DIT-Vin). To a cold anhydrous toluene suspension (40 mL) of diatomite (2.5 g) (non-activated) was added Et_3_N (1 mL, 7 µmol) dropwise a toluene solution (2 mL) of VinSi(OMe)_3_ (1.5 mL). The resulting suspension was refluxed with magnetic stirring at 110 °C for 7 h under Ar stream. The solid face was separated by decantation and washed with toluene once and alcohol (EtOH) 5–6 times. The obtained solid was oven-dried at 50 °C.

Synthesis of diatomite with covalent-grafted phosphonic groups (DIT-Vin-PA_cov_). Sample *DIT-Vin* (3.5 g) water suspension (20 mL) was mixed with H_3_PO_3_ (5 mL, 60 wt.%). The mixture was heated at 50 °C for 1 h. Then, the suspension was cooled to room temperature and mixed continuously with magnetic stirring for 5 h. The product was washed with water 9–10 times to reach pH close to the natural one. Finally, the prepared composite was air-dried in a drying oven at 50–60 °C for 10 h.

Synthesis of diatomite with non-covalent-grafted phosphonic groups (DIT-PA). The raw diatomite was activated by acidic treatment and calcinated at 440 °C for 4 h in oven. Activated diatomite was mixed with H_3_PO_3_ (5 mL) and heated at 50 °C for 1 h. The solid was filtered and washed with H_2_O to obtain the neutral pH value. Thus-prepared sample was lighter than other obtained diatomite-based materials.

Synthesis of diatomite with covalent-grafted phosphonic groups via initiator (DIT-Vin-PA_in_). The 2.16 g *DIT-Vin* (non-washed by EtOH) was mixed with 40 mL anhydrous toluene. Then, powder AIBN (0.052 g) and phosphonic acid crystalline (1.0 g) were added to the suspension and stirred magnetically fast at 80 °C for 1 h. To reaction mixture was added 0.1 g AIBN and heated at 111 °C for 17 h under Ar atmosphere. The resulting diatomite-based sample was washed with EtOH and air-dried at 60 °C overnight.

### 2.4. Characterization Methods

Powder XRD data were obtained using a PAN Analytical PW3050/60X’Pert PRO apparatus equipped with an X’Celerator detector at room temperature with automatic data acquisition (X’Pert Data Collector (v. 2.0b) software). High-resolution diffractometer was used, operating at a Ni-filtered CuKα (λ = 1.5406 Å) radiation. The diffractograms were collected at room temperature using monochromatized CuKα radiation as incident beam (40 kV–30 mA) in the 10 < 2θ degree < 80 range. XRD patterns were recorded with a step size of 0.02° at continuous scan mode, with 200 s per step.

N_2_ adsorption–desorption isotherms at 77 K were obtained by using the ASAP 2020 (Micromeritics Instrument Corp., Norcross, GA, USA) instrument. Samples were degassed at 100 °C for at least 10 h prior to obtaining 10^−3^ hPa before measurements.

*Fourier-transform* infrared spectra of obtained solids were recorded using the Spectrum Two FT-IR spectra (PerkinElmer, Waltham, MA, USA) spectrometer. The 32 successive scans were performed in the wavenumber range of 400–4000 cm^−1^ with a resolution of 4 cm^−1^. The background was subtracted from each scan to correct for atmospheric and instrumental noise. The KBr pellet technique (1:10) in an agate mortar was applied.

The scanning electron microscope (”EM, ’SM-6100 JEOL) was used to prove the structure and morphology of the samples. The energy dispersive X-ray spectra were measured in the same microscopy equipped with a Microanalysis INCA Energy 200.

The analysis of the elemental composition of the solid samples was performed using the Axios mAX X-ray Fluorescence (XRF) spectrometer (PANalytical, Almelo, The Netherlands). The samples for the measurements were prepared from the homogeneous material, which was first ground in the rotational mill made from the tungsten carbide and then dried at 120 °C. The pressed tablets were prepared for the measurements, which were made using the spectrometer WD-XRF with the wave dispersion in the 11 spectral ranges—the measurement range from sodium to uranium in the atomic number scale. The excitation source was the X-ray lamp with the rhodium anode Rh-SST-mAX with 4 kW power. The results were elaborated based on the residential program Omnian and the spectrometer calibration from the series of synthetic patterns.

Single pulse ^13^C and ^31^P MAS NMR spectra were performed with an Avance III 400WB Bruker. Spectrometer was operated at a carbon frequency of 100.61 MHz and a spinning frequency of 5 kHz. The 90° pulse width was 4.5 ms, and high-power proton decoupling was performed during recording of the spectra. The recycle delay was 20 s, and 4000 scans were added for every spectrum.

Concentrations of active functional groups in the samples were determined with two types of acid–base titrations: conductometric and pH-metric. All the conductometric titrations were measured by Consort K611 conductometer. A 20 mL aqueous suspension of 0.05 g solid was used, and suspensions were incubated overnight. Titrations were performed with 0.05 M NaOH to obtain titration curve at 25 ± 0.2 °C. The pH-metric titrations were performed by temperature-controlled И-160M apparatus and combined glass electrode. A batch of the sample with protolithic active groups (~0.1 g) was poured with 20 mL of 0.1 M NaNO_3._ The suspensions were incubated for 12 h (overnight) and titrated with 0.05 M NaOH solution.

### 2.5. Adsorption Experiments

A stock solution of U(VI) (10 mmol/L) was prepared by dissolving UO_2_(CH_3_COO)_2_·2H_2_O in deionized water. The working solution was 0.5 mmol/L. All adsorption experiments were performed in 100 mL glass conical flasks with 50 mL of U(VI) stock solution. The contact time of the sorbents with the uranium solution on a mechanical shaker at 25 °C was 10 h. For each measurement, three parallel repetitions were carried out. In order to study the effect of sorbent dose, different amounts of *DIT-Vin*, *DIT-Vin-PA_cov_*, *DIT-PA*, and *DIT-Vin-PA_in_* sorbents (0.01, 0.02, 0.05, 0.1, and 0.2 g) were added to 50 mL of 1.0 mmol/L stock solution. The effect of the pH of the solution was tested over a pH range of 2 to 10 after adding 0.5 g of the appropriate sorbent to 50 mL of a 1.0 mmol/L stock solution. For adsorption kinetics studies, 0.05 g of sorbent was weighed and mixed with 50 mL of 1.0 mmol/L U(VI) solution and then shaken for 0.5–12 h.

Adsorption isotherms were obtained by contacting 0.05 g of the appropriate sorbent with 50 mL of uranium(VI) solution, with a concentration ranging from 0.1 to 1.2 mmol/L for 10 h at 25 °C (293 K) and 50 °C (323 K). After this time, the solutions were filtered (Filtrak 390) and centrifuged (1000 rpm). The U(VI) ion concentration in the filtrates was determined spectrophotometrically (Jasco V-660, Tokyo, Japan) with Arsenazo III. Briefly, an aliquot (10 mL) of filtrate was placed in a volumetric flask (25 mL). Then, 10.0 mL of HNO_3_ (12 mol/L) and 3 mL of Arsenazo III (0.07% *w/v*) were consecutively added. The rose solution was mixed well for 30 min for complete complexation of all the U(VI) ions by the organic dye. The tested solution was consequently transferred into a 1.0 cm quartz cell, and the absorbance was measured at 665 nm.

The adsorption capacity of obtained materials was calculated as the difference between the amount of U(VI) ions in the initial solution and the amount remaining in the system after equilibration using the following equation:(1)$$q_{e}=\frac{(C_{0}-C_{e})\cdot V}{m}$$
(2)R=C0−CeC0·100%
where *q_e_* is the adsorption capacity, mmol/g; *C*_0_ and *C_e_* are the concentrations of U(VI) ions in the initial and equilibrium solutions, respectively, mmol/L*; R* is the recovery of U(VI) ions in the aqueous solution, %; *V* is the volume of the U(VI) solution, mL; and m is the weight of adsorbent, g.

### 2.6. Desorption Experiments

In order to investigate the desorption of U(VI) ions adsorbed on a particular sorbent, 0.5 g of a sample of the appropriate uranium adsorbents was weighed and then contacted for 6 h with 100 mL of HNO_3_ (pH = 3), H_2_O, 0.01 mol/L Na_2_CO_3_, 0.01 mol/L NaCl, and 0.01 mol/L Na_2_SO_4_. Then, the solutions were filtered, and the concentration of U(VI) ions in the filtrates was determined.

### 2.7. Real Water Assay

To demonstrate the applicability and reliability of the synthesized adsorbents for water treatment, environmental water and wastewater were used. Tap water, well water, natural mineral water, and sea water samples were collected in acid-leached glass bottles filtered through 0.45 μm Millipore cellulose acetate membrane filters to remove mechanical particles.

The tested water samples were found to contain other components of various origins, as shown in Table 1. The real water samples were spiked with 0.1 mg/L of U(VI) ions.

The trace concentration of metal ions in the solutions after adsorption was determined by atomic adsorption spectrometry (C-115PK Selmi) and inductively coupled plasma optical emission spectrometry (iCAP 6300, Thermo Scientific, Waltham, MA, USA).

## 3. Results and Discussion

### 3.1. Chemical Modification and Characterization of Diatomite Adsorbents

In our research, the raw diatomite material was studied before and after sulfuric acid purification. The surface morphology of the initial diatomite and washing derivate was obtained using the scale dependency of the SEM data (Figure 1).

Figure 1a indicates that the raw material exhibited highly porous disk- and rod-like shapes, with sizes of approx. 0.5–50 μm. Numerous well-developed pores can be clearly observed, indicating the high honeycomb porosity and accessible surface of diatomite as expected. It was also found that the natural mineral showed specific surface area of up to 30.0 m^2^/g. However, there were many inclusions on the surface of the raw diatomite. Most of the pores on the surface of diatomite were opened. The average diameter of the pores was observed to be approximately 50–500 nm in the peripheral area of the diatomite disk. In Figure 1b, after washing the surface of the diatomite in general, it became clean, with clearly identifiable channels. We can clearly see that the porous structure of the diatomite changed as expected after the purification process.

The XRF and EDX methods (Appendix A) are applied to determinate the chemical composition of the natural diatomite because the principal constituents of diatomite depend on the origin of the diatomaceous earth. The phase compositions of the solids obtained from the analyzed data are presented in Table 2.

As can be seen, the EDX results showed that the main components of the natural diatomite were SiO_2_ and Al_2_O_3_, with small amounts of Fe_2_O_3_, MgO, and K_2_O. Metal ions were mainly in the form of carbonate, according to the EDX data (Table 2). Additional XRF analyses showed that the chemical composition was consistent with the results obtained with the EDX analysis.

*Phosphorylation of natural diatomite.* A modification of the diatomite by phosphonic-derivative groups was carried out in several steps using covalent and non-covalent approaches (Figure 1). The following parameters were optimized for all the phosphorylation reactions: temperature, reagents concentration, reaction time, and pH.

Route I involved the preparation of a P-containing functional group with direct interaction of raw diatomite with H_3_PO_3_ (non-covalent approach) combined with calcination. In *Route II*, the natural diatomite was reacted with trimethoxy(vinyl)silane followed by a salinization reaction (covalent approach). Silanization of the diatomite is associated with the interaction of silanes with the hydroxyl groups present on the inner walls and the surface of the natural diatomite (mainly ≡ Si-OH). Two synthetic approaches were applied for the phosphorylation of the vinyl-functionalized diatomite (*DIT-Vin*). In the first route, *DIT-Vin* was reacted with the H_3_PO_3_ solution in the presence of AIBN as an initiator (*DIT-Vin-PA_in_*). In the second route, *DIT-Vin* along with heating was reacted with the H_3_PO_3_ solution (*DIT-Vin-PA_cov_*). Based on this, H_3_PO_3_ was also directly covalently immobilized on *DIT-Vin*.

To compare the immobilizing behaviours of the phosphonic groups on the diatomite support with different surface chemistries, we investigated the natural diatomite and derivative solids via various instrumental methods, such as powder XRD, solid state NMR, and FT-IR analysis.

The structures of the starting diatomite and the P-derivative solids were investigated by powder XRD (Figure 2).

The XRD pattern of natural diatomite (Figure 2) demonstrated essentially an amorphous silica phase, revealed as a “*halo*” between 16° and 26° (2θ). There were also several reflections at 17.7, 19.6, 20.8, 26.3, 27.8, 30.1, 34.5, 36.4, 39.5, 40.1, 45.6, 50.0, 60.1, etc., of crystalline impurities and reflections of quartz (2θ = 20.87, 26.66, 36.54, etc.) (Appendix A), calcite, and illite, which were present in the amount of approximately 2–3%, according to a semi-quantitative evaluation of the mineral (Table 1). The efficient immobilisation of silane on the surface of the diatomite during the silanization process (Figure 1) was confirmed by the appearance of the second “*halo*” reflection in the small angle region (up to 10°). This effect could be attributed to the increase in the amorphous silica layer after hydrolysis and corresponded to the polycondensation of the anchoring -Si(OEt)_3_ groups of the silane. It can be clearly seen that other sharp and intense diffraction peaks of the raw material were observed for the prepared organo-diatomite, indicating that the structure of the starting diatomite was not destroyed after the modification.

The chemical characterization of the surface of the diatomite-based samples with the phosphonic groups and the determination of the structure of the functional layer was performed by solid state ^29^Si and ^31^P MAS NMR spectroscopy (Figure 3 and Figure 4).

The ^29^Si MAS NMR spectrum of the natural diatomite, besides the major signals at −92.7 ppm (Q_3_ = HO−Si(OSi)_3,_ MgO−Si(OSi)_3_, or AlO−Si(OSi)_3_ sites) and −107.6 ppm (Q_4_ = [Si−(OSi)_4_] sites of quartz), showed that the peaks for *DIT-Vin*, *DIT-Vin-PA_in_*, and *DIT-Vin-PA_cov_* samples could be assigned to different Q species, i.e., geminal silanols (Q_2_, δ = −99.1 ppm), single silanols (Q_3_, δ = −102.9 ppm), and siloxane groups (Q_4_; δ = −110.2 ppm), and contained different T_n_ species (1 < n ≤ 3), where n is related to the number of siloxane bonds formed between the surface and the functional center at −60.9 and −65.1 ppm that can be assigned to T_3_ [R−Si−(OSi)_3_] and T_2_ [R_2_−Si−(OSi)_2_] groups, respectively. The effect of the silanization corresponding to the covalent Si-C bond on the surface of adsorbents indicated the presence of the polycondensed layer of organo-silane on the diatomite surface.

The magnitude of the chemical shift and the width of the ^31^P MAS NMR signal is an important tool for explaining the interactions of phosphonic acids with the matrix based on the oxide materials [27,46]. The ^31^P CP/MAS NMR spectra of the DIT-Vin sample with contact times of 0.5, 2.0, 5.0, and 10.0 ms are shown in Figure 4. Many fuzzy peaks and noises at the low times appear in the spectra of the diatomite with the physically adsorbed H_3_PO_3_ (DIT-Vin sample). The peak intensities for the ^31^P spectra changed as the time changed. The intensities for the main peaks of the phosphonic groups increased, and those for the noises decreased, as *t* increased. The observed ^31^P peaks of the DIT-Vin sample were assigned by triad configurations. Namely, these signals at 35.2 ppm, 27.9 ppm, and 19.2 ppm are believed to be related to the various types of phosphonic acid centres present in the natural material. This might have been due to the interaction between the canter bonding (-Mg-OH, ≡Si-OH, and =Al-OH) of the diatomite and H_3_PO_3_ that led to the “apparent” difference in their acidic activities. The ^31^P MAS NMR spectra were also recorded for the other P-containing diatomite samples (see Figure 5).

The ^31^P MAS NMR spectra (Figure 5) displayed intensive signals only at 28–30 ppm, indicative of the formation of uniform phosphorus environments using both covalent approached for synthesizing the diatomite samples with the immobilized phosphonic groups (*DIT-Vin-PA_cov_* and *DIT-Vin-PA_in_* samples). In the ^31^P NMR spectra of the phosphonic acids on the diatomite surfaces, there was a low shift and a splitting of the signal into two components: a low-field shift (up to 1–3 ppm) of the narrow component of the doublet due to the physically adsorbed acid molecules and signals of expanded high-field resonances (equaling 5–7 ppm), which could be attributed to the chemicals bounded with the diatomite surface phosphonates by forming the C-OP bonds [47]. The unsuccessful interaction of the phosphonic acids with the diatomite in the non-polar solvents is due to the instability of the Si-O-P and (Mg/Al)-O-P bonds [48] in contrast to the DIT-PA sample, which allowed their modification in aqueous media.

The diatomite samples before and after modification were analysed by FTIR spectroscopy. The obtained spectra are shown in Figure 6.

In the FTIR spectra of all diatomite-based samples, the vibration bands appeared at approximately 1250, 1090, 798, and 470 cm^−1^, which could be attributed to the condensed silica network (Figure 6). The intense broad band observed at 1000–1200 cm^−1^ was assigned to the stretching vibration of the Si-O-Si siloxane group framework bonds. In the spectra, only the low peak at 700 cm^−1^ represented the bending vibration of the Al–O groups; the peak at 679 cm^−1^ indicated the vibration of the Mg-O groups of the diatomite. The characteristic shoulder of the stretching vibrations of Fe-O at 570 cm^−1^ did not appear. In addition, the peaks at 2860 cm^−1^, 2930 cm^−1^, 955 cm^−1^, and 842 cm^−1^ were caused by the stretching vibration of the –C-H functional groups. The C-H bands in the FTIR spectra of the natural diatomite appeared without the chemical modification of the surface. Diatomite could contain the carbon element because it formed from the accumulation of single-celled aquatic plants of diatoms [40,41]. Carbon was the main element of the algae plant. The C-H peak was instead related to the organo-modified materials rather than to the other natural silica-based minerals [27]. The band at 3750 cm^−1^ corresponded to the single silanol groups (SiO-H), and the broad band at 3425 cm^−1^ represented the H-bonding vibration of the OH-groups from the Al-OH, Mg-OH, and Si-OH fragments. In addition, the peaks of symmetric stretching vibrations of the SiO-H bonds at 798 and 950 cm^−1^ were observed. All these characteristic peaks suggested that the tested diatomite contained mainly silica dioxide. These data are consistent with the XRD analysis.

The chemical modification of the diatomite by the vinyl functional groups was accompanied by the ν(C-H), δ(C-H), and ν(C=C) stretches. The new intensive band around 1385 cm^−1^ was assigned to the δ(C-H) bending of the vinyl groups. The *DIT-Vin-PA_in_* spectrum also did not display transmittances due to ν(SiO-H(isolated)) (3750 cm^−1^) and ν(SiO-H(vicinal)) (3660 cm^−1^), reflecting the relatively low concentration of the residual silanols and the hydrophobicity of the sample. The peak intensity of SiO-H and O-H in the modified diatomite was higher than that in the starting diatomite, possibly because the silanization reaction increased the quantity of SiO-H and OH by means of silane incorporation. The ν_s_(P-C) bands at 720–790 cm^−1^ and ν_s_(P-O) at 1100–1270 cm^−1^ could not be identified in the real spectra of organo-diatomite because of the overlap with the bands of diatomite (mainly SiO_2_ matrix) supporting the absorption region.

The evolution of the amount of active functional groups of the diatomite-based samples after various modification approaches was performed by acid–base potentiometric or conductometric titrations [49]. The shapes of the pH-metric and conductometric titration curves of the natural diatomite using the NaOH solution are shown in Figure 7.

The pH of the raw diatomite suspension in water was close to 6.2 (neutral), Figure 7. The conductivity of the aqueous suspension of diatomite in the selected range increased in an almost linear proportion to the quantity of the added NaOH solution. This effect in the conductometric curve of the raw diatomite in relation to H^+^/OH^−^ was due to the weak acid–base properties of the Si-OH or Al-OH proton-active sites on the surface of the solid [49].

Then, pH titration was used to test whether the natural diatomite was indeed capped by the protolitic-active functional groups (Figure 8). The polyfunctional surface of the obtained material made it difficult to determine the accurate concentration of the acidic centres from the potentiometric titration alone; thus, the corresponding first derivative of the titration curves was applied.

Two equivalence points were clearly defined in the integral curve of the pH-metric titration of a phosphonic acid-group-functionalized diatomite suspension with the NaOH solution (Figure 8). Such behaviour of the titration curve was indicative of the simultaneous presence of several acid groups of different strengths on the surface. Thus, the interaction of the suspension with the alkali occurred stepwise, similar to the titration of phosphonic acid in solution [49]. Furthermore, according to the titration data, the concentration of the acidic functional groups on the surface of the *DIT-Vin-PA_cov_* and *DIT-Vin-PA_in_* samples were calculated to be 2.2 mmol/g and 1.69 mmol/g, respectively, thus indicating that the adsorbents with the phosphonic acid groups were successfully synthesized. When using the covalent approach for synthesis of the P-containing adsorbents, the concentration of the functional groups was higher for the time-effective non-covalent approach (*DIT-PA* sample).

In addition, the initial value of the pH suspensions of the P-containing diatomite samples was higher than the one for the natural mineral (Figure 7), which indicated the presence of mobile ions in the solution. This could be due to the dissociation of the phosphonic acid groups. Thus, the proposed covalent immobilization of the functional phosphonic groups on the surface did not significantly change the acidic properties of H_3_PO_3_ in contrast to the previous results [49,50].

To further evaluate the effects of the phosphonic group modification approaches on the adsorption properties of the obtained materials, the starting natural and organo-diatomite samples were used for the U(VI) ions adsorption tests.

### 3.2. Effects of the Phosphonic Groups Modification of Diatomite on Its U(VI) Ion Adsorption Performance

The pH value is an important parameter influencing the adsorption process, especially by protolithic active P-containing fictionalized materials [50]. Diatomite, as a material containing mainly silica with some additive of metal oxides, was limited to pH ≥ 9.0 during dissolution in the alkaline media (≡Si-OH_(solid)_ + NaOH_(solution)_ ⇒ ≡Si-O^−^ Na^+^ + H_2_O or =Al-O-Al=_(solid)_ + NaOH ⇒ =Al-O^−^ Na^+^ + =Al-O-H). Depending on the nature of the liquid, radioactive wastes may have pH values ranging from 5.7 to 12.3, with the inorganics and radionuclides being the primary contaminants [51]. Based on these data, the tests were carried out in the pH range from 3 to 9.

Firstly, the pH effect of the U(VI) ions entrapment onto the natural diatomite and the obtained samples was evaluated, as shown in Figure 9.

It can be clearly observed that the adsorption performance of the obtained adsorbents was highly dependent on the solution pH values, and the U(VI) entrapment efficiency increased gradually with the increasing pH of the solutions (Figure 9). The removal efficiency of U(VI) was higher for the P-containing diatomite samples than for the raw material, indicating that the dominant mechanism for the U(VI) ion uptake with the modified adsorbents was achieved through an outer-sphere complexation with the functional groups. That particular adsorption behaviour can be attributed to the fact that there may exist one main interaction mechanism during the removal process. The maximum adsorption (92% and 95.5%) occurred at an initial pH near 9 (Figure 9). Hence, this pH was used in further studies. The obtained effect was particularly satisfactory because the radioactive wastewater from nuclear power plants is usually alkaline [35,51].

Additionally, it is worth noting that most of the materials reported in the literature are effective in removing uranyl ions from the neutral pH (4–5), but their efficiency drops significantly in the case of high pH values (Table 3).

Secondly, the pH initial solutions of the obtained suspensions under consideration were compared with the pH of the solutions in the equilibrium state, and the results are shown in Figure 10.

As shown in Figure 10, the pH values were significantly changed for the adsorption of the U(VI) ions by the studied adsorbents from the aqueous solutions. The decrease in pH upon adsorption of the U(VI) ions by the P-containing samples (*DIT-Vin-PA_cov_*, *DIT-PA* and *DIT-Vin-PA_in_*) was higher compared with the natural diatomite. In the case of *DIT-Vin* in the entire range of the initial pH of 3–10, the pH value in the equilibrium solutions was near 4.1, while for *DIT-Vin-PA*, this value was much lower and amounted to 3.1–3.6. This could be attributed to the fact that the phosphonic functional groups on the surfaces of *DIT-PA, DIT-Vin-PA_cov_*, and *DIT-Vin-PA_in_* were able to form stable chelates with U(VI). It is clear that the sorption process was scarcely influenced at all the pH values, suggesting that the dominant mechanism is inner-sphere surface complexation rather than outer-sphere surface complexation or ion exchange [31,50]. The highest adsorption capacity in the entire tested pH range was found for the *DIT-Vin-PA_cov_* and *DIT-Vin-PA_in_* samples.

The pH effect on the U(VI) ion sorption could be explained by the surface characteristics of the adsorbents and the speciation of the metal forms in the solute. The curves representing the change in U(VI) adsorption percentage with pH were calculated with MINTEQ 3.1 (Figure 11).

The existence of uranyl ion (UO_2_^2+^) is possible only at pH values ≤ 2.5 (Figure 11). As the solution pH was increased, the amounts of UO_2_^2+^ substantially decreased with the increasing pH due to the processes of hydrolysis and complexation. The complex products with carbonate ions, such as UO_2_CO_3_, UO_2_(CO_3_)_2_^2−^, and UO_2_(CO_3_)_3_^4−^ species, were formed. However, U(VI) ions were present in the real water supplies as anionic carbonate complexes, UO_2_(CO_3_)_2_^−2^, and UO_2_(CO_3_)_3_^−4^ at 5.0·mmol/L of HCO_3_^−^/CO_3_^2−^ ions in the real aqueous solution because of the shift in the carbon dioxide balance. Thus, under standard environmental water (pH 7–7.5) and wastewater (pH 8–8.5) conditions, uranium typically occurs in anionic carbonate forms. Hence, the solution pH is a crucial parameter for the removal of U(VI) ions.

Thus, when the pH value was lower, the surfaces of the P-containing adsorbents exhibited an acidic form, leading to the low adsorption ability of the adsorbents for the predominant positively charged forms of the U(VI) ions (Figure 11). Along with the increase in the pH value, the surface of the adsorbent became negatively charged due to the deprotonation process, so that the attraction between the adsorbent and U(VI) likely strengthened the interaction between the two. However, with further increases in the pH value, the adsorption capacity remained unchanged. The removal of the U(VI) ions likely resulted from the formation of the different uranyl hydroxide and carbonate complexes that were negatively charged, resulting in the decrease in the pH suspension at the uptake of the U(VI) ions. Such a change in pH during the adsorption process provided a large share of the ion exchange reaction with the hydronium H_3_O^+^ ions and the interaction with OH^−^ ions as competing ions for the U(VI) ions.

To investigate the maximum adsorption capacities of the adsorbents, the isotherm adsorptions of U(VI) ions were studied at pH 5.5. In addition, the influence of the temperature (293 K and 323 K) changes in the concentration of the U(VI) ions on the obtained adsorbents was investigated (Figure 12). Equilibrium experiments at the isotherm conditions are considered to be an effective process to understand the adsorption mechanisms in the purification processes [10,15].

The results of Figure 12 demonstrated that the adsorption capacity of obtained materials increased with the increasing equilibrium concentration of the U(VI) ions, progressively saturating all the studied adsorbents. The sorption characteristics of the modified samples were much higher as compared with those for the pristine diatomite. At the same time, the adsorption isotherm of the U(VI) ions on the *DIT-Vin* was σ-shaped, with an asymmetrically convex shape; thus, this was an *S2*-type isotherm according to the Giles classification [52]. In contrast, the adsorption isotherms of the U(VI) ions on the *DIT-Vin-PA_cov_* and *DIT-Vin-PA_in_* samples were *L2*-type and symmetrically convex throughout the concentration range, confirming the chemisorption mechanism removal of the analyte. A preliminary estimation of the initial part of the isotherm slopes—because these are largely determined by the affinity of the adsorbate for the adsorbent, and accordingly, the *DIT-Vin-PA_cov_* and *DIT-Vin-PA_in_* samples—showed the highest affinity toward the U(VI) ions and the lowest affinity toward the natural diatomite.

In addition, the adsorption capacities of the adsorbents at room temperature increased after the modification of the natural diatomite (Figure 12a). For instance, the adsorption capacities of the raw diatomite, *DIT-Vin*, *DIT-PA*, *DIT-Vin-PA_cov_*, and *DIT-Vin-PA_in_* toward the U(VI) ions were 0.22 mmol/g (59.4 mg/g), 1.02 mmol/g (275.4 mg/g), 0.96 mmol/g (256.83 mg/g), 1.22 mmol/g (325.7 mg/g), and 1.21 mmol/g (323.0 mg/g), respectively, indicating the highest adsorption capacities for the *DIT-Vin-PA_cov_* sample. In fact, the adsorption performance of *DIT-Vin-PA_cov_* was close to that of *DIT-Vin-PA_in_*, indicating that using an initiator during the synthetic process did not significantly increase the adsorption capacity of the adsorbents toward the U(VI) ions. The adsorption capacity of the phosphoric acid chemically modified diatomite samples was more than 5-fold greater than that on the natural diatomite, demonstrating that the silylation of the diatomite significantly increased its adsorption capacity. At the same time, the phosphoric acid activated diatomite (*DIT-PA* sample) had the advantages of availability of the reagents, simple synthesis, and good adsorption capacity. In addition, the radionuclide-saturated DIT-PA sample could be transformed into ceramics at temperatures up to 1000 °C for subsequent long-term burial.

Additionally, the maximum adsorption capacity of the samples toward the U(VI) ions reached the concentration of the functional groups of samples according to the titrations data (Figure 8). In this case, there was one phosphonic group for every uranyl ion, so the ratio was 1:1. According to the Lewis acid–base theory, uranium(VI) tends to chelate and coordinate with oxygen-containing anions to form stable coordination bonds [1,39]. According to this theory, we can conclude that the oxygen atoms in the phosphonic groups of diatomite chelated with the U(VI) ions.

On the basis of the obtained experimental data and the results presented in the literature, we could assume the possibility of complex formation of different compositions for the considered metals by analogy to the solutions, i.e., with other coordination numbers of the central atom. In this regard, using the model of the chemical reactions [50], the stability constants of some complexes were calculated. The equilibrium data used in the calculation of these constants are presented in Table 4.

From the calculations, it follows that under the conditions of the experiment, complexes of these metals in a ratio of 1:1 are formed, and at the least, their concentrations are not more than two orders of magnitude lower than the error. For the uranyl(VI) ions, the most probable was the formation of complexes of a 1:1 composition because, in the case of complexes of composition ML_2_ and ML_3_, the error was quite significant. Overall, the complex formation based on the equilibrium calculations agreed well with the isotherm adsorption data (Figure 12a).

From Figure 12b, the value of the adsorption capacity of the samples toward the U(VI) ions increased (near 10%) with the increasing solution temperature, suggesting that increasing the temperature from 298 K to 323 K favoured the adsorption. This could also confirm the noticeably greater influence of the temperature on the adsorption process for these adsorbents.

To optimize the design of an adsorption system for the target ions, the relationship between the equilibrium concentration (*C_e_*, mmol/L) and the adsorption capacity (*Q_S_*, mmol/g) could be characterized by essential isothermal models, including the Langmuir, Freundlich, and Dubinin–Radusckevich models [53,54,55]. These are the most frequently used models to describe adsorption isotherms to predict the adsorption mechanism. The Langmuir, Freundlich, and Dubinin–Radushkevich equations are expressed in linear forms:(3)$$\frac{C_{e}}{Q_{S}}=\frac{1}{Q_{max}K_{L}}+\frac{C_{e}}{Q_{max}}$$
(4)$${logQ}_{S}=logK_{F}+\frac{1}{n}logC_{e}$$
(5)$$ln\;Qs=lnq_{m}-\beta \varepsilon^{2}$$
where *C_e_* is the concentration of the analyte solution at equilibrium (mmol/L); *Q_S_* is the corresponding amount of the analyte adsorbed on the adsorbent at equilibrium (mmol/g); *Q_max_* (mmol/g) and *K_L_* (L/mmol) are constants related to the maximum adsorption capacity and energy of adsorption, respectively; *K_F_* is the Freundlich constant (mmol^1−1/n^ L^1/n^/g); *n* is a constant, which measures the intensity; *K_D-R_* is the Dubinin–Radushkevich constant (mol^2^/kJ^2^); and *ɛ* is the Polanyi element. The energy of adsorption *E* can be calculated from the formula: E = 1/(2 × K_D-R_)^0.5^.

The linearized forms and the corresponding fitting parameters of the adsorption isotherms of U(VI) ions on the obtained sorbents according to various models are listed in Appendix A and Table 5, respectively.

Studies on the parameters of the isotherms and coefficients of correlation (Table 5) show that the Langmuir and Freundlich models are the best for describing the U(VI) adsorption on the natural diatomite. From Table 5, it can be observed that the correlation coefficients for the Freundlich isotherm model for the *DIT-Vin* sample (R^2^ = 0.927) were higher compared with those for the Langmuir (R^2^ = 0.44) model. The Freundlich isotherm assumes the existence of multilayer sorption and the presence of energetically differentiated active sites on the adsorbent surface. An important parameter is the 1/n ratio, which is an indicator of the variation in free enthalpy associated with the sorption from the solution by the various components of the heterogeneous sorbent. In the case of the *DIT-Vin-PA_cov_*, *DIT-PA, DIT-Vin-PA_in_*, and raw diatomite adsorbents, when 1/n < 1, the isotherm had a convex shape, which indicated that the added U(VI) ions reacted with the centres with less and less free enthalpy. While 1/n > 1, for the *DIT-Vin* sorbent, the isotherm had an increasing concave course, from which it could be concluded that the greater amount of uranium ions on the surface increased the free enthalpy of the process. In this case, the best fit of the experimental results was obtained from the Freundlich model (R^2^ = 0.927).

The adsorption isotherms of the *DIT-PA*, *DIT-Vin-PA_cov_*, and *DIT-Vin-PA_in_* samples fit the Langmuir model well (R^2^ = 0.977, 0.99, and 0.993). The Langmuir theory has basic assumptions—the surface containing the adsorbing sites on which the adsorption occurs is a perfectly flat plane with no folds, and each site can hold at most one molecule of adsorbent (mono-layer coverage only). These assumptions suggest that the U(VI) ions were adsorbed in the form of monolayer coverage on the surface of the *DIT-PA, DIT-Vin-PA_cov_*, and *DIT-Vin-PA_in_* samples. Additionally, the maximum adsorption capacity of the samples towards the U(VI) ions was higher than the adsorption capacities of other the obtained adsorbents and the adsorbents alone in the literature (Table 3).

The Dubinin–Radushkevich isotherm equation was used to estimate the adsorption energy [55]. Table 5 shows that the values of the adsorption energy for the *DIT-Vin-PA_cov_*, *DIT-PA*, and *DIT-Vin-PA_in_* adsorbents were in the range of 8–9 kJ/mol, which may suggest a mixed (i.e., both physical and chemical) adsorption mechanism. The adsorption energy was increased with the increasing temperature for all diatomite-based adsorbents, which was characteristic for chemisorption. Chemical adsorption usually requires high activation energy and is a relatively slow process. Its speed can be increased by increasing the temperature. The heat of adsorption is of the same order as the chemical reactions and usually amounts to several kJ/mol. It decreases as the degree of coverage of the adsorbent surface increases. There were different adsorption centres in the case of the *DIT-PA* and *DIT-Vin-PA_in_* adsorbents. One of them was based on the weak van der Waals interaction of the adsorbate with the adsorbent. This was responsible for the desorption of the U(VI) ions from the surface of the sample. On the other hand, for other adsorption sites which required more adsorption energy, the adsorbent and the adsorbate interacted by a chemical reaction.

The temperature dependence of the adsorption process was associated with changes in several thermodynamic parameters, such as the standard Gibbs free energy change (ΔG°), standard enthalpy change (ΔH°), and standard entropy change (ΔS°) of adsorption, which were calculated using the following equations:ΔG° = −RT ln K_L_
ΔG° = ΔH° − ΔS° T·
where R is the gas constant (8.314 J/mol K) and T is the temperature (K).

The parameters calculated via the thermodynamic equations are shown in Appendix A and Table 6.

The values of ΔG° were found to be from −3.40 to −9.05 kJ/mol. The negative value of ΔG° indicated that the adsorption reaction was spontaneous. The Gibbs free energy decreased with the increase in the temperature, which suggested that the higher temperature may have facilitated the adsorption uranium (VI) ions on the obtained samples due to a greater driving force of the adsorption.

The positive value of ΔH° showed the endothermic nature of the adsorption process. It is well known that adsorption processes are, in most cases, exothermic [15], and generally, an exothermic adsorption process signifies either physi- or chemisorption. The positive ΔS° values displayed the affinity of adsorbents toward U(VI) ions, incarnating some structural changes.

### 3.3. Desorption Study

In order to investigate the possibility of using the obtained adsorbents in many adsorption cycles, desorption with the HNO_3_, H_2_O, Na_2_CO_3_, NaCl, and Na_2_SO_4_ solutions was performed. The samples previously saturated with U(VI) ions were then leached for 6 h using the above solutions. Then, the concentration of the uranyl ions in the eluate was determined. The desorption conditions and the results of the experiment are given in Table 7.

In the case of H_2_O and NaCl, no desorption was detected (Table 7). The highest desorption among the different leaching agents was observed for the HNO_3_ solution (pH = 3), i.e., 58–65% in the case of *DIT-Vin-PA_cov_*, *DIT-PA*, and *DIT-Vin-PA_in_* sorbents, making it possible to reuse these sorption materials. This was justified if the competition between the H^+^ and UO_2_^2+^ ions in the access to the material sorptive sites is taken into account. It is interesting that in the case of Na_2_CO_3_ and Na_2_SO_4_, 5–8% of U(VI) was desorbed, which was a result of the U(VI) ions in the presence of soluble carbonate complexes in the aqueous phase (Figure 11). The low desorption of U(VI) from the natural diatomite surface was likely a result of a higher participation of chemisorption in the overall U(VI) adsorption, than that which occurred in the case of the organo-diatomite.

These results showed that the regeneration of both natural diatomite and organo-diatomite was much less inefficient than expected. At the same time, the desorption reagents used in the tests could constitute groundwater components and did not leach out the uranium that had already been adsorbed. Thus, the most important finding of the present study was that the diatomite-supported materials tested here were also potentially a good material for the construction of the geological barriers for the radionuclides near the places that had a risk of uranium contamination.

Generally, these sorbents could be effective as a material that permanently retains uranium, and the possibility of its re-leaching into the environment is negligible.

### 3.4. Effect of Interfering Ions

Under the real conditions of the water samples, different main components and pollutants may be present at the same time. To further investigate the reaction between the P-containing adsorbents and the contaminants, the potential interferences of some metal ions on the removal effective of U(VI) ions were examined. The *DIT-Vin-PA_in_* sample was applied as the model adsorbent for this test. In these experiments, solutions of 0.1 mmol/L of the target analyte containing the interfering ions were treated by 0.05 g adsorbent according to the optimized procedures (Table 7).

Table 8 shows the tolerance limits of the interfering ions. The presence of Ca^2+^, Mg^2+^, and Zn^2+^ ions with the same concentration and different concentration gradients were investigated, and the removal percentage was basically the same as that of the solution containing only U(VI). It is speculated that Fe(III) and Cu(II), as metal ions, compete with the U(VI) ions, which may lead to their inhibition of U(VI) removal. However, the proposed adsorbents could not be used for the quantificative removal of the U(VI) ions in the environmental samples if the concentration of Fe(III) and Cu(II) ions were more than five time higher than the total concentration of the target analyte. According to the experimental results, the KSCN^−^ solution should be used as a masking agent at a level of 0.1 M (Table 8). In this case, these coexistence ions in the solution had almost no effect on the removal of the U(VI) ions.

In addition, several common anions, such as Cl^−^, SO_4_^2−^, NO_3_^−^, and F^−^, were tested. The results showed that they did not interfere at the concentrations up to 100 mg/L, whereas the U(VI) adsorption decreased at high concentrations of PO_4_^3−^. This is likely because the complex reaction between U(VI) and the dissociated anions would exert a competitive effect on the U(VI) sorption onto the adsorbents. For CO_3_^2−^, when the pH value was greater than 6, the carbonate uranyl complexes were the main forms of uranium (Figure 11), which indicated that CO_3_^2−^ had little interference on the removal of U(VI).

Therefore, the obtained materials can also display superior adsorption performance when used in practice for a mixture of multiple pollutants.

### 3.5. Treatment of Industrial Effluent and Other Types of Real Water Samples

To demonstrate the performance of the obtained adsorbents, the analyte concentration in different water samples was determined. In addition, the water samples were spiked with a standard solution of U(VI) ions (1.0 mg/L) to assess the matrix effects. The effect of the adsorbent dose on U(VI) removal from real wastewater was investigated (Figure 13).

It can be seen that the percentage of U(VI) adsorption increased with increasing *DIT-Vin-PA_in_* and *DIT-Vin-PA_cov_* dosages, and an almost complete removal (~100%) of U(VI) from the wastewater samples was achieved with 1.5 g *DIT-Vin-PA_in_* and *DIT-Vin-PA_cov_* in 1.0 L. Therefore, the optimum adsorbent dose was kept at 1.5 g/L for the adsorption experiments.

Ultimately, the uptakes were performed at optimized conditions, and the results are shown in Table 9.

The results showed that the water matrices in our present context had little effect on the water purification procedure using the *DIT-Vin-PA_in_* and *DIT-Vin-PA_cov_* adsorbents. As could be seen, the relative recoveries for the spiked water samples were in an acceptable range (88.4–99.6%). The percent relative standard deviations (RSD) were between 2.2% and 4.6%. Thus, the developed water purification procedure based on P-containing samples was successfully applied to the removal U(VI) ions in the various water samples, and satisfactory results were obtained.

## 4. Conclusions

A series of novel diatomites modified by phosphonic groups were prepared via various approaches. Covalent grafting of the diatomite with the initiator AIBN allowed us to obtain the adsorbent with the highest concentration of ethyl(phosphonic)groups (2.2 mmol/g). Therefore, the fundamental possibility of using diatomites with phosphonic groups to extract uranyl(VI) ions, as well as heavy metals, from their acidified solutions was shown. The sorbents obtained on the basis of diatomites were characterized by a very high sorption capacity relative to the U(VI) ions, many times higher than the starting material, for which the sorption was at the level of 9% and below. The adsorption process was endothermic, as evidenced by the increase in the percentage of sorption at 323 K for all sorbents. The *DIT-Vin-PA_cov_* and *DIT-Vin-PA_in_* adsorbents proved to be the most effective for uranyl ion removal. A high percentage of adsorption of 92% and 95.5% was obtained for the starting solutions at pH values of 5, 7, and 8, and 72% or 69% was obtained for pH 3. The latter result was particularly satisfactory due to the fact that radioactive wastewater from nuclear power plants is usually acidic, close to pH 3. In addition, it is worth noting that most sorbents were equally effective in removing the uranyl ions from the neutral environment, but their effectiveness dropped significantly in the case of low pH values. The maximum adsorption performance could be checked by changing the content of the phosphonic groups in the order of *DIT-Vin-PA_in_* > *DIT-PA* > *DIT-Vin-PA_cov_* > *DIT-V_in_* > *Diatomite* at 298 K. Compared with other non-covalent-modified diatomite (*DIT-PA*), the adsorbents had high adsorption capacity and removal performance, showing a potential to provide new low-cost materials for radionuclide waste disposal. The above data show that the modification of the diatomite with H_3_PO_3_ acid proved to be a very good way to obtain a high-performance sorbent for U(VI) ions in a wide pH range. The thermodynamic parameters indicated the affinity of the P-containing adsorbents toward the U(VI) ions in aqueous solutions and suggested some structural changes in the sorbents during adsorption process.

The proposed methodology can be used for the production of sorbents using P-enriched natural minerals, which enables obtaining the demanded functional materials for U(VI) ion removal in large volumes.

## Data Availability

Not applicable.

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
