# Peer review of "Optimal Synthesis of Novel Phosphonic Acid Modified Diatomite Adsorbents for Effective Removal of Uranium(VI) Ions from Aqueous Solutions"

_materials, 2023, doi:10.3390/ma16155263_

Round 1

Reviewer 1 Report

Please do a plagiarism scan - the first sentence already looks awfully familiar...

Please also do language editing...

Fig. cannot be read...needs revision

The experiments are well described. I think you need to discuss that you are only using a synthetic solution.

Qamouche et al. (https://doi.org/10.1016/j.mineng.2021.107085) provide adsorption experiments for both a synthetic solution and a real industrial waste solution showing that there are considerable differences.

So your work is useful, but it is just the beginning and in a next step an actual industrial wastewater need to be analysed. You can still publish what you have now, but you should at least discuss this.

Author Response

RESPONSE TO REVIEWERS

We are thankful for the Reviewers and Editor for the kind evaluation of our manuscript.

Response to Reviewer 1 comments:

  1. Please do a plagiarism scan - the first sentence already looks awfully familiar...

Response: Thank you for your recommendations. The manuscript has been significantly corrected.

  1. Please also do language editing... cannot be read...needs revision.

Response: The new version of manuscript has been significantly edited.

  1. The experiments are well described. I think you need to discuss that you are only using a synthetic solution.

Response: Thank you for good question. In this work, the main idea of study is focused to obtain of P-containing diatomite-based adsorbents by various synthetic routes. The resulted adsorbents were tested on a model solution with reproducible properties. Usually, real liquid radioactive wastewaters have a complex multicomponent composition, which cannot always be correctly interpreted to describe the mechanism of target radionuclide ions sorption by various sorbents.

  1. Qamouche et al. (https://doi.org/10.1016/j.mineng.2021.107085) provide adsorption experiments for both a synthetic solution and a real industrial waste solution showing that there are considerable differences.

Response: Thanks for this reference. In the work of Qamouche et al. argan nutshell sawdust from Morocco were used for extraction of uranium ions from individual solutions and wastewaters. In this case, this adsorbent has no specificity for the removal of multiply charged radionuclides, therefore, when testing the material for the extraction of uranium from real stagnant waters, the results differed significantly from studies in individual solutions.

In our work, for the extraction of U(VI), we proposed phosphorylated adsorbents that are selective to this radionuclide and have a high sorption capacity (near 300 mg/g). Of course, the quantitative recovery of the radionuclide in wastewater treatment using the obtained adsorbents can and will not be achieved, as expected higher recovery rates than when using argan nutshell with a sorption capacity of 0.93 mg/g (Qamouche et al.). Thus, the results of our study are encouraging and clearly justify further, intensified research in this direction.

  1. So, your work is useful, but it is just the beginning and in a next step an actual industrial wastewater need to be analyzed. You can still publish what you have now, but you should at least discuss this. 

Response: Our main goal was to investigate the mechanism of uranium(VI) sorption on the obtained sorbents. In the further part of our research, we plan to focus on the practical aspect and use our sorbents to treat synthetic wastewater. The preparation of simulated radioactive effluents requires consideration of many aspects. The model solution must have the correct pH and composition, including total dissolved solids, organic matter, and competing ions. It is known that uranium adsorption is greatly enhanced at low pH, while uranium speciation varies with pH, ionic strength and concentration. The pH affects the type of multinuclear complexes formed, the free uranyl cation (UO22+) is present at low pH<4, while UO2(CO3)34− is obtained above pH 8.5. However, the formation of polynuclear (UO)2CO3(OH)3 is increased above pH 6-8 with increasing uranium concentration. Awareness of these difficulties is essential to the design of advanced adsorbents to achieve the highest uranium uptake. We will continue this research in the future.

Reviewer 2 Report

Kobylinska et al. show novel phosphonic acid-modified diatomite adsorbents and their application in the removal of U(VI) ions from aqueous solutions. The work is interesting due to the applied potential and the systematic materials characterization. However, there are issues that need to be carefully addressed.

The style and the English language should be improved throughout the text.

In the introduction section, the authors have missed mentioning important materials for immobilizing radionuclides. For example, the synthetic analogs of the mineral sitinakite (TAM-5, IONSIV® IE-911, SGU-45, zeolites, micas, and many others). The authors should carefully check the available literature.

The experimental section is missing details. For example, what is the step size and time for data collection per step in the XRD?

Figure 1 caption. “……………………various magnetization………..”. What are the values of magnetization?

Figure 2. Why are there XRD peaks that are not identified?

Author Response

RESPONSE TO REVIEWERS

We are thankful for the Reviewers and Editor for the kind evaluation of our manuscript.

Response to Reviewers comments:

Reviewer 2

Kobylinska et al. show novel phosphonic acid-modified diatomite adsorbents and their application in the removal of U(VI) ions from aqueous solutions. The work is interesting due to the applied potential and the systematic materials characterization. However, there are issues that need to be carefully addressed.

  1. The style and the English language should be improved throughout the text. 

Response: The manuscript has been significantly edited.

  1. In the introduction section, the authors have missed mentioning important materials for immobilizing radionuclides. For example, the synthetic analogs of the mineral sitinakite (TAM-5, IONSIV® IE-911, SGU-45, zeolites, micas, and many others). The authors should carefully check the available literature.

Response: It was corrected in revised manuscript.

  1. The experimental section is missing details. For example, what is the step size and time for data collection per step in the XRD?

Response: Thank you. The detailed information about the step size and time for data collection per step in the XRD measurement added to the experimental part.

  1. Figure 1 caption. “……………………various magnetization………..”. What are the values of magnetization?

Response: The morphology of diatomite samples is studied with SEM images using magnification 1500x and 1000x.

  1. Figure 2. Why are there XRD peaks that are not identified? 

Response: We have identified most of the peaks in the XRD patterns of the obtained samples. At the same time, the main idea of X-ray diffraction studies of samples is evaluated the effect of chemical immobilization on the structure of natural diatomite.

Round 2

Reviewer 1 Report

Sorry, I do not see my comments addressed. I will ask the editor to also consider adding more reviewers if you do not feel my comments are helpful.

Author Response

RESPONSE TO REVIEWERS

We are thankful for the Reviewers and Editor for the kind evaluation of our manuscript.

Response to Reviewers comments:

Reviewer 1

  1. Please do a plagiarism scan - the first sentence already looks awfully familiar...

Response: Thank you for your recommendations. The manuscript has been significantly corrected.

  1. Please also do language editing... cannot be read...needs revision.

Response: The new version of manuscript has been significantly edited.

  1. The experiments are well described. I think you need to discuss that you are only using a synthetic solution.

Response: Thank you for good question. In this work, the main idea of study is focused to obtain of P-containing diatomite-based adsorbents by various synthetic routes. The resulted adsorbents were tested on a model solution with reproducible properties. Usually, real liquid radioactive wastewaters have a complex multicomponent composition, which cannot always be correctly interpreted to describe the mechanism of target radionuclide ions sorption by various sorbents.

Finally, the results of real wastewater analysis have been added in the new version of manuscript.

  1. Qamouche et al. (https://doi.org/10.1016/j.mineng.2021.107085) provide adsorption experiments for both a synthetic solution and a real industrial waste solution showing that there are considerable differences.

Response: Thanks for this reference. In the work of Qamouche et al. argan nutshell sawdust from Morocco were used for extraction of uranium ions from individual solutions and wastewaters. In this case, this adsorbent has no specificity for the removal of multiply charged radionuclides, therefore, when testing the material for the extraction of uranium from real stagnant waters, the results differed significantly from studies in individual solutions.

In our work, for the extraction of U(VI), we proposed phosphorylated adsorbents that are selective to this radionuclide and have a high sorption capacity (near 300 mg/g). Of course, the quantitative recovery of the radionuclide in wastewater treatment using the obtained adsorbents can and will not be achieved, as expected higher recovery rates than when using argan nutshell with a sorption capacity of 0.93 mg/g (Qamouche et al.). Thus, the results of our study are encouraging and clearly justify further, intensified research in this direction.

  1. So, your work is useful, but it is just the beginning and in a next step an actual industrial wastewater need to be analyzed. You can still publish what you have now, but you should at least discuss this. 

Response: It was corrected. The results of real tap, well, river and industrial wastewaters analysis have been added in the revised version of manuscript.

Reviewer 2

Kobylinska et al. show novel phosphonic acid-modified diatomite adsorbents and their application in the removal of U(VI) ions from aqueous solutions. The work is interesting due to the applied potential and the systematic materials characterization. However, there are issues that need to be carefully addressed.

  1. The style and the English language should be improved throughout the text. 

Response: The manuscript has been significantly edited.

  1. In the introduction section, the authors have missed mentioning important materials for immobilizing radionuclides. For example, the synthetic analogs of the mineral sitinakite (TAM-5, IONSIV® IE-911, SGU-45, zeolites, micas, and many others). The authors should carefully check the available literature.

Response: It was corrected in revised manuscript.

  1. The experimental section is missing details. For example, what is the step size and time for data collection per step in the XRD?

Response: Thank you. The detailed information about the step size and time for data collection per step in the XRD measurement added to the experimental part.

  1. Figure 1 caption. “……………………various magnetization………..”. What are the values of magnetization?

Response: The morphology of diatomite samples is studied with SEM images using magnification 1500x and 1000x.

  1. Figure 2. Why are there XRD peaks that are not identified? 

Response: We have identified most of the peaks in the XRD patterns of the obtained samples. At the same time, the main idea of X-ray diffraction studies of samples is evaluated the effect of chemical immobilization on the structure of natural diatomite.
